# Regularized $M$-estimators with nonconvexity: Statistical and algorithmic theory for local optima

**Po-Ling Loh**
Department of Statistics
University of California, Berkeley
Berkeley, CA 94720
ploh@berkeley.edu

**Martin J. Wainwright**
Departments of Statistics and EECS
University of California, Berkeley
Berkeley, CA 94720
wainwrig@stat.berkeley.edu

## Abstract

We establish theoretical results concerning local optima of regularized $M$-estimators, where both loss and penalty functions are allowed to be nonconvex. Our results show that as long as the loss satisfies restricted strong convexity and the penalty satisfies suitable regularity conditions, *any local optimum* of the composite objective lies within statistical precision of the true parameter vector. Our theory covers a broad class of nonconvex objective functions, including corrected versions of the Lasso for errors-in-variables linear models and regression in generalized linear models using nonconvex regularizers such as SCAD and MCP. On the optimization side, we show that a simple adaptation of composite gradient descent may be used to compute a global optimum up to the statistical precision $\epsilon_{\text{stat}}$ in $\log(1/\epsilon_{\text{stat}})$ iterations, the fastest possible rate for any first-order method. We provide simulations to illustrate the sharpness of our theoretical predictions.

## 1 Introduction

Optimization of nonconvex functions is known to be computationally intractable in general [11, 12]. Unlike convex functions, nonconvex functions may possess local optima that are not global optima, and standard iterative methods such as gradient descent and coordinate descent are only guaranteed to converge to local optima. Although statistical results regarding nonconvex $M$-estimation often only provide guarantees about the accuracy of *global* optima, it is observed empirically that the *local* optima obtained by various estimation algorithms seem to be well-behaved.

In this paper, we study the question of whether it is possible to certify "good" behavior, in both a statistical and computational sense, for various nonconvex $M$-estimators. On the statistical level, we provide an abstract result, applicable to a broad class of (potentially nonconvex) $M$-estimators, which bounds the distance between *any local optimum* and the unique minimum of the population risk. Although local optima of nonconvex objectives may not coincide with global optima, our theory shows that any local optimum is essentially as good as a global optimum from a statistical perspective. The class of $M$-estimators covered by our theory includes the modified Lasso as a special case, but our results are much stronger than those implied by previous work [6].

In addition to nonconvex loss functions, our theory also applies to nonconvex regularizers, shedding new light on a long line of recent work involving the nonconvex SCAD and MCP regularizers [3, 2, 13, 14]. Various methods have been proposed for optimizing convex loss functions with nonconvex penalties [3, 4, 15], but these methods are only guaranteed to generate local optima of the composite objective, which have not been proven to be well-behaved. In contrast, our work provides a set of regularity conditions under which *all local optima* are guaranteed to lie within a small ball of the population-level minimum, ensuring that standard methods such as projected and composite gradient descent [10] are sufficient for obtaining estimators that lie within statistical error of the

truth. In fact, we establish that under suitable conditions, a modified form of composite gradient descent only requires $\log(1/\epsilon_{\text{stat}})$ iterations to obtain a solution that is accurate up to the statistical precision $\epsilon_{\text{stat}}$.

**Notation.** For functions $f(n)$ and $g(n)$, we write $f(n) \precsim g(n)$ to mean that $f(n) \leq cg(n)$ for some universal constant $c \in (0, \infty)$, and similarly, $f(n) \succsim g(n)$ when $f(n) \geq c'g(n)$ for some universal constant $c' \in (0, \infty)$. We write $f(n) \asymp g(n)$ when $f(n) \precsim g(n)$ and $f(n) \succsim g(n)$ hold simultaneously. For a function $h : \mathbb{R}^p \to \mathbb{R}$, we write $\nabla h$ to denote a gradient or subgradient, if it exists. Finally, for $q, r > 0$, let $\mathbb{B}_q(r)$ denote the $\ell_q$-ball of radius $r$ centered around 0.

## 2 Problem formulation

In this section, we develop some general theory for regularized $M$-estimators. We first establish notation, then discuss assumptions for nonconvex regularizers and losses studied in our paper.

### 2.1 Background

Given a collection of $n$ samples $Z_1^n = \{Z_1, \ldots, Z_n\}$, drawn from a marginal distribution $\mathbb{P}$ over a space $\mathcal{Z}$, consider a loss function $\mathcal{L}_n : \mathbb{R}^p \times (\mathcal{Z})^n \to \mathbb{R}$. The value $\mathcal{L}_n(\beta; Z_1^n)$ serves as a measure of the "fit" between a parameter vector $\beta \in \mathbb{R}^p$ and the observed data. This empirical loss function should be viewed as a surrogate to the *population risk function* $\mathcal{L} : \mathbb{R}^p \to \mathbb{R}$, given by

$$\mathcal{L}(\beta) := \mathbb{E}_Z\big[\mathcal{L}_n(\beta; Z_1^n)\big].$$

Our goal is to estimate the parameter vector $\beta^* := \arg \min_{\beta \in \mathbb{R}^p} \mathcal{L}(\beta)$ that minimizes the population risk, assumed to be unique.

To this end, we consider a regularized $M$-estimator of the form

$$\widehat{\beta} \in \arg \min_{g(\beta) \leq R} \{\mathcal{L}_n(\beta; Z_1^n) + \rho_\lambda(\beta)\}, \tag{1}$$

where $\rho_\lambda : \mathbb{R}^p \to \mathbb{R}$ is a *regularizer*, depending on a tuning parameter $\lambda > 0$, which serves to enforce a certain type of structure on the solution. In all cases, we consider regularizers that are separable across coordinates, and with a slight abuse of notation, we write $\rho_\lambda(\beta) = \sum_{j=1}^p \rho_\lambda(\beta_j)$.

Our theory allows for possible nonconvexity in *both* the loss function $\mathcal{L}_n$ and the regularizer $\rho_\lambda$. Due to this potential nonconvexity, our $M$-estimator also includes a side constraint $g : \mathbb{R}^p \to \mathbb{R}_+$, which we require to be a convex function satisfying the lower bound $g(\beta) \geq \|\beta\|_1$, for all $\beta \in \mathbb{R}^p$. Consequently, any feasible point for the optimization problem (1) satisfies the constraint $\|\beta\|_1 \leq R$, and as long as the empirical loss and regularizer are continuous, the Weierstrass extreme value theorem guarantees that a global minimum $\widehat{\beta}$ exists.

### 2.2 Nonconvex regularizers

We now state and discuss conditions on the regularizer, defined in terms of $\rho_\lambda : \mathbb{R} \to \mathbb{R}$.

**Assumption 1.**

(i) The function $\rho_\lambda$ satisfies $\rho_\lambda(0) = 0$ and is symmetric around zero (i.e., $\rho_\lambda(t) = \rho_\lambda(-t)$ for all $t \in \mathbb{R}$).

(ii) On the nonnegative real line, the function $\rho_\lambda$ is nondecreasing.

(iii) For $t > 0$, the function $t \mapsto \frac{\rho_\lambda(t)}{t}$ is nonincreasing in $t$.

(iv) The function $\rho_\lambda$ is differentiable for all $t \neq 0$ and subdifferentiable at $t = 0$, with nonzero subgradients at $t = 0$ bounded by $\lambda L$.

(v) There exists $\mu > 0$ such that $\rho_{\lambda, \mu}(t) := \rho_\lambda(t) + \mu t^2$ is convex.

Many regularizers that are commonly used in practice satisfy Assumption 1, including the $\ell_1$-norm, $\rho_\lambda(\beta) = \|\beta\|_1$, and the following commonly used nonconvex regularizers:

**SCAD penalty:** This penalty, due to Fan and Li [3], takes the form

$$\rho_\lambda(t) := \begin{cases} \lambda|t|, & \text{for } |t| \leq \lambda, \\ -(t^2 - 2a\lambda|t| + \lambda^2)/(2(a-1)), & \text{for } \lambda < |t| \leq a\lambda, \\ (a+1)\lambda^2/2, & \text{for } |t| > a\lambda, \end{cases} \tag{2}$$

where $a > 2$ is a fixed parameter. Assumption 1 holds with $L = 1$ and $\mu = \frac{1}{a-1}$.

**MCP regularizer:** This penalty, due to Zhang [13], takes the form

$$\rho_\lambda(t) := \text{sign}(t)\,\lambda \cdot \int_0^{|t|} \left(1 - \frac{z}{\lambda b}\right)_+ dz, \tag{3}$$

where $b > 0$ is a fixed parameter. Assumption 1 holds with $L = 1$ and $\mu = \frac{1}{b}$.

### 2.3 Nonconvex loss functions and restricted strong convexity

Throughout this paper, we require the loss function $\mathcal{L}_n$ to be differentiable, but we do not require it to be convex. Instead, we impose a weaker condition known as restricted strong convexity (RSC). Such conditions have been discussed in previous literature [9, 1], and involve a lower bound on the remainder in the first-order Taylor expansion of $\mathcal{L}_n$. In particular, our main statistical result is based on the following RSC condition:

$$\langle \nabla \mathcal{L}_n(\beta^* + \Delta) - \nabla \mathcal{L}_n(\beta^*),\, \Delta \rangle \geq \begin{cases} \alpha_1 \|\Delta\|_2^2 - \tau_1 \dfrac{\log p}{n} \|\Delta\|_1^2, & \forall \|\Delta\|_2 \leq 1, \quad (4a) \\[2mm] \alpha_2 \|\Delta\|_2 - \tau_2 \sqrt{\dfrac{\log p}{n}} \|\Delta\|_1, & \forall \|\Delta\|_2 \geq 1, \quad (4b) \end{cases}$$

where the $\alpha_j$'s are strictly positive constants and the $\tau_j$'s are nonnegative constants.

To understand this condition, note that if $\mathcal{L}_n$ were actually strongly convex, then both these RSC inequalities would hold with $\alpha_1 = \alpha_2 > 0$ and $\tau_1 = \tau_2 = 0$. However, in the high-dimensional setting ($p \gg n$), the empirical loss $\mathcal{L}_n$ can never be strongly convex, but the RSC condition may still hold with strictly positive $(\alpha_j, \tau_j)$. On the other hand, if $\mathcal{L}_n$ is convex (but not strongly convex), the left-hand expression in inequality (4) is always nonnegative, so inequalities (4a) and (4b) hold trivially for $\frac{\|\Delta\|_1}{\|\Delta\|_2} \geq \sqrt{\frac{\alpha_1 n}{\tau_1 \log p}}$ and $\frac{\|\Delta\|_1}{\|\Delta\|_2} \geq \frac{\alpha_2}{\tau_2}\sqrt{\frac{n}{\log p}}$, respectively. Hence, the RSC inequalities only enforce a type of strong convexity condition over a cone set of the form $\left\{\frac{\|\Delta\|_1}{\|\Delta\|_2} \leq c\sqrt{\frac{n}{\log p}}\right\}$.

## 3 Statistical guarantees and consequences

We now turn to our main statistical guarantees and some consequences for various statistical models. Our theory applies to any vector $\widetilde{\beta} \in \mathbb{R}^p$ that satisfies the *first-order necessary conditions* to be a local minimum of the program (1):

$$\langle \nabla \mathcal{L}_n(\widetilde{\beta}) + \nabla \rho_\lambda(\widetilde{\beta}),\, \beta - \widetilde{\beta} \rangle \geq 0, \qquad \text{for all feasible } \beta \in \mathbb{R}^p. \tag{5}$$

When $\widetilde{\beta}$ lies in the interior of the constraint set, condition (5) is the usual zero-subgradient condition.

### 3.1 Main statistical results

Our main theorem is deterministic in nature, and specifies conditions on the regularizer, loss function, and parameters, which guarantee that any local optimum $\widetilde{\beta}$ lies close to the target vector $\beta^* = \arg\min_{\beta \in \mathbb{R}^p} \mathcal{L}(\beta)$. Corresponding probabilistic results will be derived in subsequent sections. For proofs and more detailed discussion of the results contained in this paper, see the technical report [7].

**Theorem 1.** Suppose the regularizer $\rho_\lambda$ satisfies Assumption 1, $\mathcal{L}_n$ satisfies the RSC conditions (4) with $\alpha_1 > \mu$, and $\beta^*$ is feasible for the objective. Consider any choice of $\lambda$ such that

$$\frac{2}{L} \cdot \max\left\{\|\nabla \mathcal{L}_n(\beta^*)\|_\infty,\ \alpha_2 \sqrt{\frac{\log p}{n}}\right\} \ \leq\ \lambda \ \leq\ \frac{\alpha_2}{6RL}, \tag{6}$$

and suppose $n \geq \frac{16R^2 \max(\tau_1^2, \tau_2^2)}{\alpha_2^2} \log p$. Then any vector $\widetilde{\beta}$ satisfying the first-order necessary conditions (5) satisfies the error bounds

$$\|\widetilde{\beta} - \beta^*\|_2 \leq \frac{7\lambda L \sqrt{k}}{4(\alpha_1 - \mu)}, \qquad \text{and} \qquad \|\widetilde{\beta} - \beta^*\|_1 \leq \frac{56\lambda L k}{4(\alpha_1 - \mu)}, \tag{7}$$

where $k = \|\beta^*\|_0$.

From the bound (7), note that the squared $\ell_2$-error grows proportionally with $k$, the number of nonzeros in the target parameter, and with $\lambda^2$. As will be clarified in the following sections, choosing $\lambda$ proportional to $\sqrt{\frac{\log p}{n}}$ and $R$ proportional to $\frac{1}{\lambda}$ will satisfy the requirements of Theorem 1 w.h.p. for many statistical models, in which case we have a squared $\ell_2$-error that scales as $\frac{k \log p}{n}$, as expected.

**Remark 1.** It is worthwhile to discuss the quantity $\alpha_1 - \mu$ appearing in the denominator of the bound in Theorem 1. Recall that $\alpha_1$ measures the level of curvature of the loss function $\mathcal{L}_n$, while $\mu$ measures the level of nonconvexity of the penalty $\rho_\lambda$. Intuitively, the two quantities should play opposing roles in our result: Larger values of $\mu$ correspond to more severe nonconvexity of the penalty, resulting in worse behavior of the overall objective (1), whereas larger values of $\alpha_1$ correspond to more (restricted) curvature of the loss, leading to better behavior.

We now develop corollaries for various nonconvex loss functions and regularizers of interest.

## 3.2 Corrected linear regression

We begin by considering the case of high-dimensional linear regression with systematically corrupted observations. Recall that in the framework of ordinary linear regression, we have the model

$$y_i = \underbrace{\langle \beta^*, x_i \rangle}_{\sum_{j=1}^p \beta_j^* x_{ij}} + \epsilon_i, \qquad \text{for } i = 1, \ldots, n, \tag{8}$$

where $\beta^* \in \mathbb{R}^p$ is the unknown parameter vector and $\{(x_i, y_i)\}_{i=1}^n$ are observations. Following Loh and Wainwright [6], assume we instead observe pairs $\{(z_i, y_i)\}_{i=1}^n$, where the $z_i$'s are systematically corrupted versions of the corresponding $x_i$'s. Some examples include the following:

(a) *Additive noise:* Observe $z_i = x_i + w_i$, where $w_i \perp\!\!\!\perp x_i$, $\mathbb{E}[w_i] = 0$, and $\text{cov}[w_i] = \Sigma_w$.

(b) *Missing data:* For $\vartheta \in [0, 1)$, observe $z_i \in \mathbb{R}^p$ such that for each component $j$, we independently observe $z_{ij} = x_{ij}$ with probability $1 - \vartheta$, and $z_{ij} = *$ with probability $\vartheta$.

We use the population and empirical loss functions

$$\mathcal{L}(\beta) = \frac{1}{2}\beta^T \Sigma_x \beta - \beta^{*T} \Sigma_x \beta, \qquad \text{and} \qquad \mathcal{L}_n(\beta) = \frac{1}{2}\beta^T \widehat{\Gamma} \beta - \widehat{\gamma}^T \beta, \tag{9}$$

where $(\widehat{\Gamma}, \widehat{\gamma})$ are estimators for $(\Sigma_x, \Sigma_x \beta^*)$ depending on $\{(z_i, y_i)\}_{i=1}^n$. Then $\beta^* = \arg\min_\beta \mathcal{L}(\beta)$. From the formulation (1), the corrected linear regression estimator is given by

$$\widehat{\beta} \in \arg\min_{g(\beta) \leq R} \left\{ \frac{1}{2}\beta^T \widehat{\Gamma} \beta - \widehat{\gamma}^T \beta + \rho_\lambda(\beta) \right\}. \tag{10}$$

We now state a corollary in the case of additive noise (model (a)), where we take

$$\widehat{\Gamma} = \frac{Z^T Z}{n} - \Sigma_w, \qquad \text{and} \qquad \widehat{\gamma} = \frac{Z^T y}{n}. \tag{11}$$

When $p \gg n$, the matrix $\widehat{\Gamma}$ in equation (11) is always negative-definite, so the empirical loss function $\mathcal{L}_n$ previously defined (9) is nonconvex. Other choices of $\widehat{\Gamma}$ are applicable to missing data (model (b)), and also lead to nonconvex programs [6].

**Corollary 1.** Suppose we have i.i.d. observations $\{(z_i, y_i)\}_{i=1}^n$ from a corrupted linear model with sub-Gaussian additive noise. Suppose $(\lambda, R)$ are chosen such that $\beta^*$ is feasible and

$$c\sqrt{\frac{\log p}{n}} \leq \lambda \leq \frac{c'}{R}.$$

Then given a sample size $n \geq C \max\{R^2, k\} \log p$, any local optimum $\widetilde{\beta}$ of the nonconvex program (10) satisfies the estimation error bounds

$$\|\widetilde{\beta} - \beta^*\|_2 \leq \frac{c_0 \lambda \sqrt{k}}{\lambda_{\min}(\Sigma_x) - 2\mu}, \qquad \text{and} \qquad \|\widetilde{\beta} - \beta^*\|_1 \leq \frac{c_0' \lambda k}{\lambda_{\min}(\Sigma_x) - 2\mu},$$

with probability at least $1 - c_1 \exp(-c_2 \log p)$, where $\|\beta^*\|_0 = k$.

**Remark 2.** When $\rho_\lambda(\beta) = \lambda\|\beta\|_1$ and $g(\beta) = \|\beta\|_1$, taking $\lambda \asymp \sqrt{\frac{\log p}{n}}$ and $R = b_0\sqrt{k}$ for some constant $b_0 \geq \|\beta^*\|_2$ yields the required scaling $n \gtrsim k \log p$. Hence, the bounds in Corollary 1 agree with bounds in Theorem 1 of Loh and Wainwright [6]. Note, however, that the latter results are stated only for a *global minimum* $\widehat{\beta}$ of the program (10), whereas Corollary 1 is a much stronger result holding for *any local minimum* $\widetilde{\beta}$. Theorem 2 of our earlier paper [6] provides an indirect route for establishing similar bounds on $\|\widetilde{\beta} - \beta^*\|_1$ and $\|\widetilde{\beta} - \beta^*\|_2$, since the projected gradient descent algorithm may become stuck in local minima. In contrast, our argument here does not rely on an algorithmic proof and applies to a more general class of (possibly nonconvex) penalties.

Corollary 1 also has important consequences in the case where pairs $\{(x_i, y_i)\}_{i=1}^n$ from the linear model (8) are observed without corruption and $\rho_\lambda$ is nonconvex. Then the empirical loss $\mathcal{L}_n$ is equivalent to the least-squares loss, modulo a constant factor. Much existing work [3, 14] only establishes statistical consistency of *global* minima and then provides specialized algorithms for obtaining specific local optima that are provably close to global optima. In contrast, our results demonstrate that *any* optimization algorithm converging to a local optimum suffices.

### 3.3 Generalized linear models

Moving beyond linear regression, we now consider the case where observations are drawn from a generalized linear model (GLM). Recall that a GLM is characterized by the conditional distribution

$$\mathbb{P}(y_i \mid x_i, \beta, \sigma) = \exp\left\{\frac{y_i\langle \beta,\, x_i\rangle - \psi(x_i^T \beta)}{c(\sigma)}\right\},$$

where $\sigma > 0$ is a scale parameter and $\psi$ is the cumulant function. By standard properties of exponential families [8, 5], we have

$$\psi'(x_i^T \beta) = \mathbb{E}[y_i \mid x_i, \beta, \sigma].$$

In our analysis, we assume there exists $\alpha_u > 0$ such that $\psi''(t) \leq \alpha_u$ for all $t \in \mathbb{R}$. This boundedness assumption holds in various settings, including linear regression, logistic regression, and multinomial regression. The bound is required to establish both statistical consistency results in the present section and fast global convergence guarantees for our optimization algorithms in Section 4.

We will assume that $\beta^*$ is sparse and optimize the penalized maximum likelihood program

$$\widehat{\beta} \in \arg\min_{g(\beta) \leq R}\left\{\frac{1}{n}\sum_{i=1}^n \left(\psi(x_i^T\beta) - y_i x_i^T\beta\right) + \rho_\lambda(\beta)\right\}. \tag{12}$$

We then have the following corollary:

**Corollary 2.** Suppose we have i.i.d. observations $\{(x_i, y_i)\}_{i=1}^n$ from a GLM, where the $x_i$'s are sub-Gaussian. Suppose $(\lambda, R)$ are chosen such that $\beta^*$ is feasible and

$$c\sqrt{\frac{\log p}{n}} \leq \lambda \leq \frac{c'}{R}.$$

Given a sample size $n \geq CR^2 \log p$, any local optimum $\widetilde{\beta}$ of the nonconvex program (12) satisfies

$$\|\widetilde{\beta} - \beta^*\|_2 \leq \frac{c_0 \lambda \sqrt{k}}{\lambda_{\min}(\Sigma_x) - 2\mu}, \qquad \text{and} \qquad \|\widetilde{\beta} - \beta^*\|_1 \leq \frac{c_0' \lambda k}{\lambda_{\min}(\Sigma_x) - 2\mu},$$

with probability at least $1 - c_1 \exp(-c_2 \log p)$, where $\|\beta^*\|_0 = k$.

# 4 Optimization algorithm

We now describe how a version of composite gradient descent may be applied to efficiently optimize the nonconvex program (1). We focus on a version of the optimization problem with the side function

$$g_{\lambda,\mu}(\beta) := \frac{1}{\lambda}\Big\{\rho_\lambda(\beta) + \mu\|\beta\|_2^2\Big\}, \tag{13}$$

which is convex by Assumption 1. We may then write the program (1) as

$$\widehat{\beta} \in \arg\min_{g_{\lambda,\mu}(\beta)\leq R}\Big\{\underbrace{\big(\mathcal{L}_n(\beta) - \mu\|\beta\|_2^2\big)}_{\bar{\mathcal{L}}_n} + \lambda g_{\lambda,\mu}(\beta)\Big\}. \tag{14}$$

The objective function then decomposes nicely into a sum of a differentiable but nonconvex function and a possibly nonsmooth but convex penalty. Applied to the representation (14), the composite gradient descent procedure of Nesterov [10] produces a sequence of iterates $\{\beta^t\}_{t=0}^\infty$ via the updates

$$\beta^{t+1} \in \arg\min_{g_{\lambda,\mu}(\beta)\leq R}\left\{\frac{1}{2}\left\|\beta - \left(\beta^t - \frac{\nabla\bar{\mathcal{L}}_n(\beta^t)}{\eta}\right)\right\|_2^2 + \frac{\lambda}{\eta}g_{\lambda,\mu}(\beta)\right\}, \tag{15}$$

where $\frac{1}{\eta}$ is the stepsize. Define the *Taylor error* around $\beta_2$ in the direction $\beta_1 - \beta_2$ by

$$\mathcal{T}(\beta_1,\beta_2) := \mathcal{L}_n(\beta_1) - \mathcal{L}_n(\beta_2) - \langle\nabla\mathcal{L}_n(\beta_2),\, \beta_1 - \beta_2\rangle. \tag{16}$$

For all vectors $\beta_2 \in \mathbb{B}_2(3) \cap \mathbb{B}_1(R)$, we require the following form of restricted strong convexity:

$$\mathcal{T}(\beta_1,\beta_2) \geq \begin{cases} \alpha_1\|\beta_1 - \beta_2\|_2^2 - \tau_1\dfrac{\log p}{n}\|\beta_1 - \beta_2\|_1^2, & \forall\|\beta_1 - \beta_2\|_2 \leq 3, \quad (17a) \\[2mm] \alpha_2\|\beta_1 - \beta_2\|_2 - \tau_2\sqrt{\dfrac{\log p}{n}}\|\beta_1 - \beta_2\|_1, & \forall\|\beta_1 - \beta_2\|_2 \geq 3. \quad (17b) \end{cases}$$

The conditions (17) are similar but not identical to the earlier RSC conditions (4). The main difference is that we now require the Taylor difference to be bounded below uniformly over $\beta_2 \in \mathbb{B}_2(3) \cap \mathbb{B}_1(R)$, as opposed to for a fixed $\beta_2 = \beta^*$. We also assume an upper bound:

$$\mathcal{T}(\beta_1,\beta_2) \leq \alpha_3\|\beta_1 - \beta_2\|_2^2 + \tau_3\frac{\log p}{n}\|\beta_1 - \beta_2\|_1^2, \qquad \text{for all } \beta_1,\beta_2 \in \mathbb{R}^p, \tag{18}$$

a condition referred to as *restricted smoothness* in past work [1]. Throughout this section, we assume $\alpha_i > \mu$ for all $i$, where $\mu$ is the coefficient ensuring the convexity of the function $g_{\lambda,\mu}$ from equation (13). Furthermore, we define $\alpha = \min\{\alpha_1,\alpha_2\}$ and $\tau = \max\{\tau_1,\tau_2,\tau_3\}$.

The following theorem applies to any population loss function $\mathcal{L}$ for which the population minimizer $\beta^*$ is $k$-sparse and $\|\beta^*\|_2 \leq 1$, and under the scaling $n > Ck\log p$, for a constant $C$ depending on the $\alpha_i$'s and $\tau_i$'s. We show that the composite gradient updates (15) exhibit a type of *globally geometric convergence* in terms of the quantity

$$\kappa := \frac{1 - \frac{\alpha-\mu}{4\eta} + \varphi(n,p,k)}{1 - \varphi(n,p,k)}, \qquad \text{where} \quad \varphi(n,p,k) := \frac{128\tau k\frac{\log p}{n}}{\alpha - \mu}. \tag{19}$$

Under the stated scaling on the sample size, we are guaranteed that $\kappa \in (0,1)$. Let

$$T^*(\delta) := \frac{2\log\left(\frac{\phi(\beta^0)-\phi(\widehat{\beta})}{\delta^2}\right)}{\log(1/\kappa)} + \left(1 + \frac{\log 2}{\log(1/\kappa)}\right)\log\log\left(\frac{\lambda RL}{\delta^2}\right), \tag{20}$$

where $\phi(\beta) := \mathcal{L}_n(\beta) + \rho_\lambda(\beta)$, and define $\epsilon_{\text{stat}} := \|\widehat{\beta} - \beta^*\|_2$.

**Theorem 2.** Suppose $\mathcal{L}_n$ satisfies the RSC/RSM conditions (17) and (18), and suppose $\rho_\lambda$ satisfies Assumption 1. Suppose $\widehat{\beta}$ is any global minimum of the program (14), with

$$R\sqrt{\frac{\log p}{n}} \leq c, \qquad \text{and} \qquad \lambda \geq \frac{4}{L}\cdot\max\left\{\|\nabla\mathcal{L}_n(\beta^*)\|_\infty,\, \tau\sqrt{\frac{\log p}{n}}\right\}.$$

Then for any stepsize $\eta \geq 2\cdot\max\{\alpha_3 - \mu,\, \mu\}$ and tolerance $\delta^2 \geq \frac{c\epsilon_{\text{stat}}^2}{1-\kappa}$, we have

$$\|\beta^t - \widehat{\beta}\|_2^2 \leq \frac{2}{\alpha-\mu}\left(\delta^2 + \frac{\delta^4}{\tau} + 128\tau\frac{k\log p}{n}\epsilon_{\text{stat}}^2\right), \qquad \forall t \geq T^*(\delta). \tag{21}$$

**Remark 3.** Note that for the optimal choice of tolerance parameter $\delta \asymp \epsilon_{\text{stat}}$, the bound in inequality (21) takes the form $\frac{c\epsilon_{\text{stat}}^2}{\alpha - \mu}$, meaning successive iterates are guaranteed to converge to a region within statistical accuracy of the true global optimum $\widehat{\beta}$. Combining Theorems 1 and 2, we have

$$\max\left\{\|\beta^t - \widehat{\beta}\|_2, \ \|\beta^t - \beta^*\|_2\right\} = \mathcal{O}\left(\sqrt{\frac{k \log p}{n}}\right), \qquad \forall t \geq T(c'\epsilon_{\text{stat}}).$$

## 5   Simulations

In this section, we report the results of simulations for two versions of the loss function $\mathcal{L}_n$, corresponding to linear and logistic regression, and three penalty functions: Lasso, SCAD, and MCP. In all cases, we chose regularization parameters $R = \frac{1.1}{\lambda} \cdot \rho_\lambda(\beta^*)$ and $\lambda = \sqrt{\frac{\log p}{n}}$.

**Linear regression:**   In the case of linear regression, we simulated covariates corrupted by additive noise according to the mechanism described in Section 3.2, giving the estimator

$$\widehat{\beta} \in \arg \min_{g_{\lambda, \mu}(\beta) \leq R} \left\{ \frac{1}{2}\beta^T \left( \frac{X^T X}{n} - \Sigma_w \right) \beta - \frac{y^T Z}{n}\beta + \rho_\lambda(\beta) \right\}. \tag{22}$$

We generated i.i.d. samples $x_i \sim N(0, I)$ and $\epsilon_i \sim N(0, (0.1)^2)$, and set $\Sigma_w = (0.2)^2 I$.

**Logistic regression:**   In the case of logistic regression, we generated i.i.d. samples $x_i \sim N(0, I)$. Since $\psi(t) = \log(1 + \exp(t))$, the program (12) becomes

$$\widehat{\beta} \in \arg \min_{g_{\lambda, \mu}(\beta) \leq R} \left\{ \frac{1}{n}\sum_{i=1}^{n} \{\log(1 + \exp(\langle \beta, \, x_i \rangle)) - y_i \langle \beta, \, x_i \rangle\} + \rho_\lambda(\beta) \right\}. \tag{23}$$

We optimized the programs (22) and (23) using the composite gradient updates (15). Figure 1 shows the results of corrected linear regression with Lasso, SCAD, and MCP regularizers for three different problem sizes $p$. In each case, $\beta^*$ is a $k$-sparse vector with $k = \lfloor \sqrt{p} \rfloor$, where the nonzero entries were generated from a normal distribution and the vector was then rescaled so $\|\beta^*\|_2 = 1$. As predicted by Theorem 1, the curves corresponding to the same penalty function stack up nicely when the estimation error $\|\widehat{\beta} - \beta^*\|_2$ is plotted against the rescaled sample size $\frac{n}{k \log p}$, and the $\ell_2$-error decreases to zero as the number of samples increases, showing that the estimators (22) and (23) are statistically consistent. We chose the parameter $a = 3.7$ for SCAD and $b = 3.5$ for MCP.

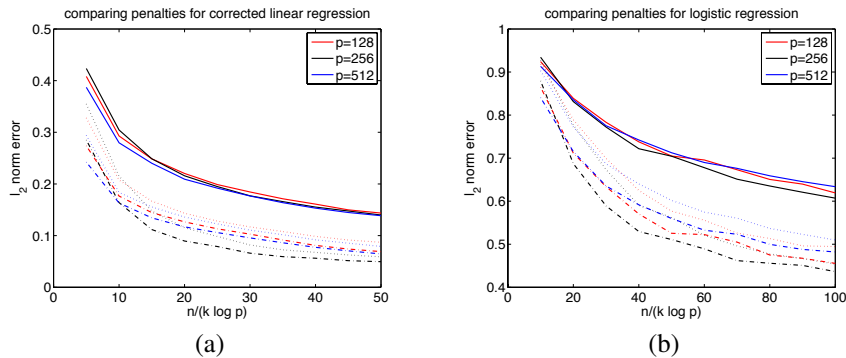

**Figure 1.** Plots showing statistical consistency of (a) linear and (b) logistic regression with Lasso, SCAD, and MCP. Each point represents an average over 20 trials. The estimation error $\|\widehat{\beta} - \beta^*\|_2$ is plotted against the rescaled sample size $\frac{n}{k \log p}$. Lasso, SCAD, and MCP results are represented by solid, dotted, and dashed lines, respectively.

The simulations in Figure 2 depict the optimization-theoretic conclusions of Theorem 2. Each panel shows two different families of curves, corresponding to statistical error (red) and optimization error

(blue). The vertical axis measures the $\ell_2$-error on a log scale, while the horizontal axis tracks the iteration number. The curves were obtained by running composite gradient descent from 10 random starting points. We used $p = 128$, $k = \lfloor\sqrt{p}\rfloor$, and $n = \lfloor 20k \log p \rfloor$. As predicted by our theory, the optimization error decreases at a linear rate until it falls to the level of statistical error. Panels (b) and (c) provide simulations for two values of the SCAD parameter $a$; the larger choice $a = 3.7$ corresponds to a higher level of curvature and produces a tighter cluster of local optima.

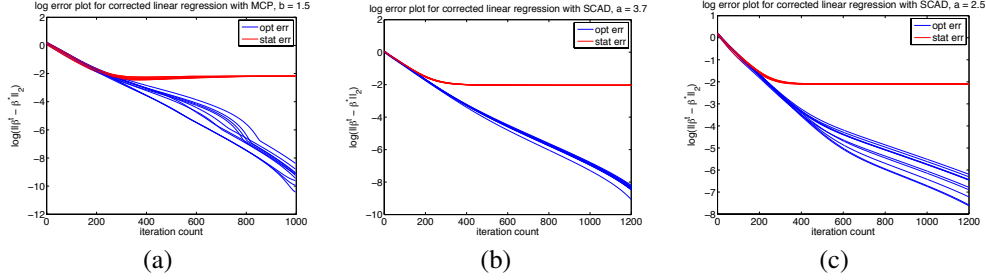

**Figure 2.** Plots illustrating linear rates of convergence for corrected linear regression with MCP and SCAD. Red lines depict statistical error $\log\left(\|\widehat{\beta} - \beta^*\|_2\right)$ and blue lines depict optimization error $\log\left(\|\beta^t - \widehat{\beta}\|_2\right)$. As predicted by Theorem 2, the optimization error decreases linearly up to statistical accuracy. Each plot shows the solution trajectory for 10 initializations of composite gradient descent. Panel (a) shows results for MCP; panels (b) and (c) show results for SCAD with different values of $a$.

Figure 3 provides analogous results to Figure 2 for logistic regression, using $p = 64$, $k = \lfloor\sqrt{p}\rfloor$, and $n = \lfloor 20k \log p \rfloor$. The plot shows solution trajectories for 20 different initializations of composite gradient descent. Again, the log optimization error decreases at a linear rate up to the level of statistical error, as predicted by Theorem 2. Whereas the convex Lasso penalty yields a unique local/global optimum $\widehat{\beta}$, SCAD and MCP produce multiple local optima.

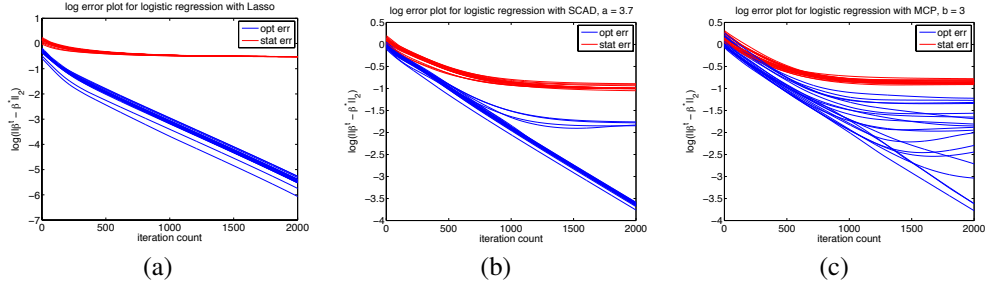

**Figure 3.** Plots showing linear rates of convergence on a log scale for logistic regression. Red lines depict statistical error and blue lines depict optimization error. (a) Lasso penalty. (b) SCAD penalty. (c) MCP. Each plot shows the solution trajectory for 20 initializations of composite gradient descent.

# 6 Discussion

We have analyzed theoretical properties of local optima of regularized $M$-estimators, where both the loss and penalty function are allowed to be nonconvex. Our results are the first to establish that *all local optima* of such nonconvex problems are close to the truth, implying that any optimization method guaranteed to converge to a local optimum will provide statistically consistent solutions. We show that a variant of composite gradient descent may be used to obtain near-global optima in linear time, and verify our theoretical results with simulations.

### Acknowledgments

PL acknowledges support from a Hertz Foundation Fellowship and an NSF Graduate Research Fellowship. MJW and PL were also partially supported by grants NSF-DMS-0907632 and AFOSR-09NL184. The authors thank the anonymous reviewers for helpful feedback.

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
