[Reviews · NeurIPS 2013]

Submitted by Assigned_Reviewer_4

The paper establishes parameter error bounds for a class of regularized m-estimators that are non-convex. The paper combines the work of 15,12, and 1 and applies these tools to the non-convex case. The primary observation seems to be that although the objective is non-convex, restricted strong convexity is enough to ensure small parameter error. The results also cover the case of nonconvex regularizers by "moving" the non-convexity to the objective.


For those not familiar with the existing results for non-convex regularized estimators, a little bit of comparison to existing results would help set the stage for the paper's contributions. What do existing theoretical results establish and how do they compare? I thought that Ref 25 offered similar statistical guarantees. Is the benefit of the current paper that all local minima are close to the truth, so we can avoid the two-step procedures previously used?

On the whole, the results are clearly presented and many illustrating examples are given. The paper is extremely well-written and clear. I strongly recommend the acceptance of this paper.

Summary: This paper is well-written and extremely clear. The results seem to be quite comprehensive and covers a wide variety of problems. This paper considerably improves the previous results on non-convex estimation. I strongly recommend the acceptance of this paper.

Submitted by Assigned_Reviewer_5

This paper establishes theoretical results for non-convex regularized M-estimators. In particular, it shows the bounds in L2 and L1 norm between any local optimum and the population optimum under the restricted strong convexity condition. Different from previous works, the statistical precision is established for all local optima. It further shows the linear convergence rate in optimization when a specialized composite gradient descent scheme is adopted. This is a very nice theoretical work which explores the properties of non-convex M-estimators from from statistical and optimization aspects. Some detailed comments are as follows:

(1) There are quite a few other non-convex penalties that have been proposed recently in addition to SCAD and MCP. For example l_q norm with 0 < q < 1, Log-Sum Penalty [1], Capped-L_1 penalty [2], etc. I am wondering, say for L_q norm penalty (or other ones), will Assumption 1 (5 conditions) be satisfied ?

(2) Could the authors provide a little bit more intuition why two separated RSC is adopted here ?

(3) It might be a typo on Line 383 ( please correct me if it is not). Since ||b^t-b^*||_2^2 < O(k log p / n *epsilon_{stat}^2). When epsilon_{stat} is on the order of O(\sqrt{k log p / n}), we have max(||b^t-\hat{b}||_2, ||b^t-b^*||_2)=O(k log(p)/n)

(4) The statistical precision is not defined in the main text, it should be ||\hat{b}-b^*||^2 or its expectation?

(5) Typo: In Appendix Eq. 72, it should be $\nabla \bar{L}_n$ instead of $\nabla L_n$.

[1] Candes, E.J., Wakin, M.B., and Boyd, S.P. Enhancing sparsity by reweighted ℓ1 minimization. Journal of Fourier Analysis and Applications, 14(5):877–905, 2008.

[2] Zhang, T. Multi-stage convex relaxation for feature selection. Bernoulli, 2012.
Summary: This paper provides nice theoretical results for regularized M-estimators with nonconvexity, on both statistical and optimization aspects.

Submitted by Assigned_Reviewer_7

summary:

This paper extends a previous line of work on analysis of the effectiveness of estimators based on non-convex optimization; in particular, it provides statistical guarantees for a class of separable non-convex functions: those that are a sum of a loss function that is non-convex but has RSC, and regularizers which are potentially non-convex themselves. For such settings the paper shows the consistency of any local (as opposed to global, or special) minimum.

clarity:

The paper is clearly written, except for

(a) the issue of the side-function g. Except for a brief discussion, there is not much to go by in trying to understand the purpose of this function. A better discussion of this would enhance the paper. Also maybe synthetic experiments to see whether this is needed in "practice".

(b) it is casually mentioned that generalizations to other "decomposable" regularizers is possible. doing so would significantly increase the applicability of this work, e.g. extending to settings like low-rank matrices etc.

originality:

The reviewer feels the paper has limited originality. In particular, a lot of the proof borrows from the previous work of Loh and Wainwright; this fact is highlighted, for example, by the fact that the result holds for very specific regularizers: those which are separable into a sum of terms, one for each coordinate (like the \ell_1 norm is).

Similarly, another generalization - to non-convex functions with RSC, instead of functions that arise from "corrected LASSO" in Loh-Wainwright - is not fleshed out much. Do these represent a straightforward generalization (along the lines of Negahban et al), or a substantial new advance ?

significance:

The significance of this paper is not immediately clear to the reviewer. In particular, while it is clear that a result about all local minima is more powerful than just a result for global minima or special initializations, it is not clear what are the settings where all the assumptions would hold.

For example, the paper seems to be focused on sparse recovery type settings, given that the regularizer is separable and has other "ell_1 - like" properties. The authors give two examples which satisfy the assumptions, but these are not widely popular in the literature. What about, for example, the more popular example of \rho(t) = |t|^p for some p less than 1 ?

Also, are there interesting/natural loss functions that are non-convex, but still have RSC and to whom the results of this a paper can be applied ? The paper seem to give no examples in this direction.
Summary: Interesting results, but concern of limited originality and significance.
Author Feedback

Author rebuttal: We thank the reviewers for their careful reading of the paper and constructive criticism, and are happy to incorporate your suggestions into the revision.

Reviewer 4:

We are glad you enjoyed our paper. To clarify, the central contribution of our paper is not limited to nonconvex loss functions satisfying RSC, but also covers families of nonconvex regularizers. Whereas the analysis of [12] could be extended naturally to general losses satisfying RSC, we have worked harder in the present paper to consider cases where the regularizer is nonconvex.

You are correct that another key point is that our paper makes sweeping statements about local optima of the nonconvex objective. Although [25] gives similar statistical guarantees, those results only concern *global* optima, necessitating a separate analysis to show that special optimization procedures converge to appropriate optima. In contrast, our paper proves that all *local* optima are in well-behaved. This distinction is important in appreciating the novelty of our paper, since it is the first to make general theoretical statements about all local optima.

Reviewer 5:

Your summary in the first paragraph is accurate, and you raise good points. Regarding (1), we agree that it would be desirable to prove statements about local optima for other nonconvex regularizers. We have results concerning the capped l_1 penalty, but we omitted them due to space limitations. What makes capped l_1 different from l_q and log-sum is that the latter two penalties have gradient tending to infinity around the origin, violating Assumption 1(iv). The infinite subgradient at 0 actually implies that \betatil = 0 is *always* a local optimum, so it cannot be true that all local optima are vanishingly close to the truth. However, a more careful analysis of specific optimization algorithms might show that the algorithms never converge to undesirable points such as the origin.

Regarding question (2), there are two important distinctions. First, regarding the difference between eqs. (4) and (22): while eqs. (22) concern arbitrary pairs (\beta_1, \beta_2), eqs. (4) fix \beta_2 = \beta^*. Indeed, the statistical results only require curvature to be controlled around \beta^*, while the optimization results apply RSC to general vector pairs. Second, in terms of having separate inequalities (a) and (b), note that although (b) follows from (a) when the loss function is convex (cf. Lemma 5), (a) generally cannot hold over the full domain R^p. For instance, in GLMs, (a) translates into lower-bounding \psi'' by a positive constant; in some settings of interest (e.g., logistic regression), \psi'' converges to 0 as the modulus of the argument increases. (See also the discussion on p. 10 of [1].)

For (3), line 383 is correct in its current form: In eq. (26), consider \delta = O(\sqrt{k log p / n}) = \epsilon_stat and \tau = O(1). In settings of interest where k log p / n < 1, the square of the optimization error will be bounded by k log p / n.

In (4), thanks for pointing out our mistake. We should have defined \epsilon_stat = ||\hat{\beta} - \beta^*||_2. Similarly, thanks for pointing out the typo in (5).

Reviewer 7:

Thanks for your close reading and thorough comments. We wish to respond directly to the question of originality/significance. Although our paper is certainly inspired by [12], our proof techniques are substantially different and provide a novel contribution to the theory (rather than being a "straightforward generalization"). Since we are interested in proving broad statements about all local optima, the key ingredient of our proofs is the first-order subgradient condition (5); in sharp contrast, related past work (e.g., [12], [15], [25]) is based on a "basic inequality" satisfied by the optimal solution. (The basic inequality holds for global optima, but *not* for local optima.) Given this key difference, all subsequent inequalities must be stated on the level of subgradients, which is by no means a straightforward extension. Additional complications arise in the optimization proofs because of the nonconvexity of the regularizer, which was not studied in [12], [1], or [16]. Hence, although Corollary 1 of our paper is implied by [12], our arguments are much more direct and generally applicable. (See Remark 2.)

In terms of applicability of the assumptions, see our response to reviewer 5 regarding the l_p penalty. Based on a literature search, l_p is not necessarily more "popular" than SCAD/MCP; these penalties are extremely widely used in practice. Moreover, many regularizers used in practice are separable. As for nonconvex loss functions satisfying RSC, the corrected Lasso is one example discussed in the paper. We have also been studying other nonconvex losses arising from regression with corrupted observations and believe RSC will hold in those cases; we agree that such examples would improve the relevance of the paper. We also agree that a more extensive discussion of the consequences of our results for low-rank matrices would make the exposition more complete, but we omitted such corollaries due to space limitations.

Finally, a few comments on the role of the side function. Note that this constraint is *necessary* to ensure that the program (1) is well-posed when the loss is nonconvex. For instance, consider the corrected Lasso in (17), when the regularizer is l_1. Setting \beta_n = c_n * \beta_0, where \beta_0 is an eigenvector corresponding to a negative eigenvalue of \hat{\Gamma}, and taking c_n -> \infty, the objective function decreases without bound. Allowing for nonconvexity in the regularizer only worsens the situation. Hence, we need an extra side constraint in order to guarantee that a global optimum of (1) exists. As you point out, an interesting direction of research would be to see whether the extra constraint is needed in the optimization algorithms, since removing the constraint would simplify the optimization steps.